# Physico-Chemical, In Vitro and Ex Vivo Characterization of Meloxicam Potassium-Cyclodextrin Nanospheres

**DOI:** 10.3390/pharmaceutics13111883

**Published:** 2021-11-06

**Authors:** Patrícia Varga, Rita Ambrus, Piroska Szabó-Révész, Dávid Kókai, Katalin Burián, Zsolt Bella, Ferenc Fenyvesi, Csilla Bartos

**Affiliations:** 1Faculty of Pharmacy, Institute of Pharmaceutical Technology and Regulatory Affairs, University of Szeged, 6720 Szeged, Hungary; varga.patricia@szte.hu (P.V.); ambrus.rita@szte.hu (R.A.); revesz@pharm.u-szeged.hu (P.S.-R.); 2Department of Medical Microbiology, Albert Szent-Györgyi Medical School, University of Szeged, 6720 Szeged, Hungary; kokai.david@med.u-szeged.hu (D.K.); burian.katalin@med.u-szeged.hu (K.B.); 3Department of Oto-Rhino-Laryngology and Head-Neck Surgery, University of Szeged, 6725 Szeged, Hungary; dr.bella.zsolt@gmail.com; 4Department of Pharmaceutical Technology, Faculty of Pharmacy, University of Debrecen, 4032 Debrecen, Hungary; fenyvesi.ferenc@pharm.unideb.hu

**Keywords:** alpha-cyclodextrin, hydroxypropyl-β-cyclodextrin, meloxicam potassium, nano spray drying, nasal drug delivery

## Abstract

Nasal drug delivery has many beneficial properties, such as avoiding the first pass metabolism and rapid onset of action. However, the limited residence time on the mucosa and limited absorption of certain molecules make the use of various excipients necessary to achieve high bioavailability. The application of mucoadhesive polymers can increase the contact time with the nasal mucosa, and permeation enhancers can enhance the absorption of the drug. We aimed to produce nanoparticles containing meloxicam potassium (MEL-P) by spray drying intended for nasal application. Various cyclodextrins (hydroxypropyl-β-cyclodextrin, α-cyclodextrin) and biocompatible polymers (hyaluronic acid, poly(vinylalcohol)) were used as excipients to increase the permeation of the drug and to prepare mucoadhesive products. Physico-chemical, in vitro and ex vivo biopharmaceutical characterization of the formulations were performed. As a result of spray drying, mucoadhesive nanospheres (average particle size <1 µm) were prepared which contained amorphous MEL-P. Cyclodextrin-MEL-P complexes were formed and the applied excipients increased the in vitro and ex vivo permeability of MEL-P. The highest amount of MEL-P permeated from the α-cyclodextrin-based poly(vinylalcohol)-containing samples in vitro (209 μg/cm^2^) and ex vivo (1.47 μg/mm^2^) as well. After further optimization, the resulting formulations may be promising for eliciting a rapid analgesic effect through the nasal route.

## 1. Introduction

Nasal administration of different active pharmaceutical ingredients (APIs) is an investigated area, since it is possible not to just treat local pathological conditions, but to reach the systemic circulation or the central nervous system through the nasal epithelia because of its high vascularization and large surface area [1]. The absorbed drugs avoid the first pass hepatic metabolism and a rapid onset of action may occur [2]. Despite the many favorable properties, there are some limitations that need to be overcome for the success of drug absorption. Due to the mucociliary clearance, the nasal fluid renews every 15–20 min which results in short residence time of the API, and the low permeability of the mucosa is also a hindering factor [3,4]. By selecting the appropriate excipients—that are mucoadhesive (e.g., chitosan, hyaluronic acid, Carbopols [5]), so the contact time of the API is extended, or have permeability enhancing features (e.g., cyclodextrins, chitosan, phospholipids [6]) by their interaction with the epithelial tight junctions—these undesirable factors can be eliminated.

Nasal drops, sprays and ointments are well known in the market, while nasal powders are not so common, although they have some advantageous properties over the other formulations. Nasal powders tend to be more difficult to eliminate from the nasal mucosa allowing a longer contact time for the API; moreover, because of their low moisture content, powders have an improved stability and there is no need to use preservatives—which are often irritative—during their production [7]. 

Spray drying is one of the most widespread, well-known bottom-up processes in the pharmaceutical industry to produce particles with controlled size and morphology in one step for, e.g., pulmonary, nasal or parenteral administration [8]. It allows for the preparation of powders even with the use of heat-sensitive compounds and the resulting products are usually amorphous because of their rapid solidification [9]. With the adjustment of the process parameters, spherical particles can be obtained which results in improved powder flowability, and the low moisture content of the products enhances their stability [10,11,12]. Nano spray drying is an effective way to increase the bioavailability of drugs, since nanoparticles have a high specific surface area compared to their size, which results in better dissolution and higher absorption rate [13]. It makes the incorporation of the APIs in polymers feasible as well, thus controlled drug release can be achieved [14,15]. BÜCHI Nano Spray Drier B-90 is a laboratory scale instrument with low sample volume requirements that produces sub-micron particles with a size range of 0.2–5 µm [16]. 

Cyclodextrins (CDs) are cyclic oligosaccharides that are famous for their capability to entrap drug molecules by forming inclusion complexes. Spray drying is a suitable method for obtaining such complexes, which may be used to increase the solubility and the dissolution of the APIs and may have drastically different properties than the original compounds. CDs can increase the permeability of the mucosal membranes, so an improved drug absorption can be achieved, and they can reduce the irritant effect of the APIs as well, so their application may be favorable in nasal formulations [17,18,19,20]. 

Hyaluronic acid (HA) and poly(vinylalcohol) (PVA) are swellable, biocompatible and biodegradable polymers that are commonly used in drug delivery. HA is a natural polysaccharide that has mucoadhesive and permeability enhancing features, so it can be used in nasal formulations for the success of drug delivery [21]. PVA is a synthetic polymer that is often used as an additive to decrease the cohesion between the spray dried particles, so that they stay separated from each other [22]. These hydrophilic polymers were chosen in order to produce potentially mucoadhesive products.

Meloxicam (MEL) is a non-steroidal anti-inflammatory drug with poor water solubility. In the therapeutic field, it is used to treat different joint diseases and it could be used to relieve acute pain [23]. Its effect on the use of opioids in the case of post-orthopedic surgery patients was studied by administering it intravenously [24]. In addition, its delivery through alternative routes—such as transdermal and nasal routes—has been recently investigated in vitro and in vivo [25,26,27]. Meloxicam potassium monohydrate (MEL-P) is the salt form of MEL that was developed by Egis Ltd. (Budapest, Hungary) and has a higher aqueous solubility (MEL-P: 13.1 mg/mL in water at 25 °C; MEL: 4.4 μg/mL in water at 25 °C). However, with different techniques, e.g., spray drying, forming inclusion complexes, incorporating into polymer matrices, its permeability and bioavailability can be further increased [28,29].

The aim of our work was to prepare MEL-P-containing nanoparticles by spray drying using various CDs and polymers to enhance the permeability of the API and to make it suitable to deliver it to the systemic circulation through the nasal epithelium to enhance analgesia or relieve acute pain. To the best of our knowledge, this combination of the materials produced by spray drying has not been investigated under nasal conditions so far. The physico-chemical characterization, mucoadhesivity investigations, in vitro and ex vivo permeability and cell cytotoxicity studies were carried out.

## 2. Materials and Methods

### 2.1. Materials

MEL-P was obtained from Egis Ltd. (Budapest, Hungary). (2-Hydroxy)-propyl-β-cyclodextrin (HPβCD) and α-cyclodextrin (αCD) were from Cyclolab Ltd. (Budapest, Hungary). Hyaluronic acid (sodium salt) (HA) was from Contipro Biotech (Dolní Dobrouč, Czech Republic), poly(vinylalcohol) (PVA) and mucin (from porcine stomach, type II) were from Sigma-Aldrich (Sigma-Aldrich Co. LLC, St. Louis, MO, USA).

### 2.2. Methods

#### 2.2.1. Preparation of the Spray Dried Samples

The solutions for nano spray drying were prepared by dissolving 1:1 mol/mol ratio of MEL-P and cyclodextrin (HPβCD or αCD) using 10 mL of distilled water as the solvent for the polymer-free samples. For the PVA-containing samples, 10 mg of PVA was added to the solutions in addition to the aforementioned compounds, and for the HA-containing samples, 5 mg of HA was dissolved in the MEL-P-cyclodextrin solutions (Table 1). BÜCHI Nano Spray Dryer B-90 HP (BÜCHI Labortechnik AG, Flawil, Switzerland) (Figure 1) was used for the production of the samples. The following parameters were applied for the process: inlet air temperature: 80 °C, pump: 20%, aspirator capacity: 100%, compressed air flow: 130 L·h^−^^1^.

#### 2.2.2. Scanning Electron Microscopy (SEM)

The size, shape and surface morphology of the spray dried particles were visualized by SEM (Hitachi S4700, Hitachi Scientific Ltd., Tokyo, Japan). Under an argon atmosphere, after sputter-coating the samples with gold-palladium in a high-vacuum evaporator, they were examined at 10 kV and 10 μA. The air pressure was 1.3–13 MPa. The size of the particles was measured by ImageJ program. From each sample, 100 particles were measured.

#### 2.2.3. Differential Scanning Calorimetry (DSC)

Mettler Toledo DSC 821^e^ system and STAR^e^ program V9.1 (Mettler Toledo Inc., Schwerzenbach, Switzerland) were used to implement the thermal analysis. Approximately 2–5 mg of samples in sealed aluminum pans were heated from 25 °C to 300 °C applying 10 °C·min^−1^ heating rate under a constant argon flow of 10 L·h^−1^. Physical mixtures (PMs) of MEL-P, cyclodextrins, HA and PVA in the same ratio as the spray dried samples contained were mixed in a Turbula mixer (Turbula WAB, Systems Schatz, Muttenz, Switzerland) at 50 rpm for 10 min and were applied as control samples.

#### 2.2.4. X-ray Powder Diffraction (XRPD)

To examine the physical state of MEL-P in the samples, XRPD was performed with a Bruker D8 Advance diffractometer (Bruker AXS GmbH, Karlsruhe, Germany) with Cu K λI radiation (λ = 1.5406 Å). The samples were scanned at 40 kV and 40 mA with an angular range of 3–40° 2θ. Si was used to calibrate the instrument. As controls, the PMs of MEL-P, cyclodextrins, HA and PVA were applied in the same ratio as the spray-dried samples contained were mixed in a Turbula mixer (Turbula WAB, Systems Schatz, Switzerland) at 50 rpm for 10 min. DIFFRACTPLUS EVA software was used to perform the manipulations: Kα2-stripping, background removal and smoothing. 

#### 2.2.5. Fourier-Transformed Infrared Spectroscopy (FT-IR)

The interactions between MEL-P and the excipients were investigated by the AVATAR330 FT-IR spectrometer (Thermo Nicolet, Unicam Hungary Ltd., Budapest, Hungary) in the interval of 400–4000 cm^−1^, at an optical resolution of 4 cm^−^^1^. Samples were grounded and compressed into pastilles at 10 t with 0.15 g of KBr.

#### 2.2.6. Mucoadhesivity

The potential mucoadhesivity of the samples was estimated by the displacement of powders on the tilted surface of agar–mucin and—as controls—pure agar gels, using a protocol proposed in the literature [30]. Briefly, a hot solution of 2% agar with or without 2% mucin in phosphate buffer pH 6.4 was poured into a petri dish and left for gelation overnight. The gels were stored at 32 °C before the test. The 7.5 mg MEL-P-containing samples were placed on top of the gels in a spot with a diameter of approximately 10 mm. At the beginning of the investigation, the petri dishes were leaned at an angle of 45° and the displacement of powder samples was measured against time. All measurements were conducted in triplicate. 

#### 2.2.7. In Vitro and Ex Vivo Permeability Studies 

A modified horizontal diffusion model (Figure 2) was applied to study the in vitro and ex vivo permeability of MEL-P [31]. This apparatus simulated the nasal conditions. The 7.5 mg of MEL-P-containing samples were added to the donor phase, which was 9 mL of SNES of pH 5.6 (represented the nasal fluid). Nine microliters of pH = 7.4 phosphate buffer—corresponding to the pH of the blood—was used as the acceptor phase. The temperature of the phases was 32 °C (Thermo Haake C10-P5, Sigma Aldrich Co.) and the rotation rate of the stir-bars was set to 300 rpm. 

For the in vitro tests, the two chambers of the apparatus were divided by an artificial membrane (Whatman^TM^ regenerated cellulose membrane filter with 0.45 μm pores) that was soaked in isopropyl myristate for 30 min before the investigation. It modeled the lipophilic mucosa between the phases. For the ex vivo measurements, the permeability test was performed on human nasal mucosa (mucoperiostium) in the case of the formulations with the highest in vitro permeability. The pieces of the nasal mucosa for primary study were collected during daily clinical routine nasal and sinus surgeries (septoplasty, FESS) under general or local anesthesia. The surgical field was infiltrated with 1% Lidocain-Tonogen local injection and the mucosa was lifted from its base with a raspatorium or Cottle elevator. Transport from the operating room was performed in physiological saline.

The amount of MEL-P diffused to the acceptor phase was determined spectrophotometrically at 364 nm in real time with an AvaLight DH-S-BAL spectrophotometer (AVANTES, Apeldoorn, The Netherlands). Each measurement was carried out in triplicate. The flux was determined at 15 min and the permeation enhancement ratios for the in vitro measurements were calculated based on the following equations (Equations (1) and (2)) [32]:(1)$$P_{\mathrm{app}}=\frac{Q}{A\cdot c\cdot t}$$
where P_app_ is the apparent permeability coefficient (cm/s), Q is the total amount permeated throughout the incubation time (μg), A is the diffusion area of the artificial membrane (cm^2^), c is the initial concentration of the drug in the donor compartment (μg/cm^3^), and t is the total time of the experiment (s) and
(2)$$R=\frac{P_{\mathrm{app}}\left(\mathrm{sample}\right)}{P_{\mathrm{app}}\left(\mathrm{control}\right)}$$
where R is the permeation enhancement ratio and P_app_ (control) is the apparent permeability coefficient (cm/s) of MEL-P.

#### 2.2.8. In Vitro Cytotoxicity Measurements

Mitochondrial activity as a measure of cell viability was performed by MTT assay in 96-well cell culture microplates using RPMI 2650 cells (human nasal septum epithelial squamous carcinoma cells, obtained from American Type Culture Collection, ATCC, Manassas, VA, USA). RPMI 2650 cells were seeded at a density of 4 × 10^4^ cells/well. First, serial dilution was made of the following stock concentrations: 4.11 mg/mL HPβCD_MEL-P_PVA_spd or 2.99 mg/mL αCD_MEL-P_PVA_spd or 5 µg/mL Lipopolysaccharide (LPS; ThermoFisher Scientific, Waltham, MA, USA), then the cells were incubated at 37 °C for 24 h. Later, 20 μL of thiazolyl blue tetrazolium bromide (MTT; Sigma-Aldrich, St. Louis, MO, USA) was added to each well. After an additional incubation at 37 °C for 4 h, sodium dodecyl sulfate (Sigma-Aldrich, St. Louis, MO, USA) solution (10% in 0.01 M HCI) was added and were incubated overnight. Cytotoxicity of the compounds was then determined by measuring the OD at 550 nm (ref. 630 nm) with EZ READ 400 ELISA reader (Biochrom, Cambridge, UK). The assay was repeated four times for each concentration.

#### 2.2.9. Examination of the Anti-Inflammatory Effect of Compounds in In Vitro Experiments

RPMI 2650 cells were seeded in 6-well plate at a density of 1 × 10^6^ cells/well and treated with the highest noncytotoxic concentration either with 1.03 mg/mL HPβCD_MEL-P_PVA_spd and 5 µg/mL LPS or 0.38 mg/mL αCD_MEL-P_PVA_spd and 5 µg/mL LPS or 5 µg/mL LPS or left untreated. During the experiment the LPS was used as a positive control.

#### 2.2.10. Total RNA Extraction and cDNA Synthesis

After 24 h treatment of RPMI 2650 cells with the compounds, RNA was extracted using the TRI reagent (Sigma-Aldrich, St. Louis, MO, USA) according to the manufacturer’s protocol. Subsequently, 1 µg of total RNA was reverse transcribed using Maxima Reverse Transcriptase (ThermoFisher Scientific, Waltham, MA, USA) according to the manufacturer’s instructions.

#### 2.2.11. qPCR Amplification of IL-6, COX-2, IL-1b, Actb

IL-6 is a cytokine and plays a major role in the inflammation, among various cell type, epithelial cell are also excreting IL-6 [34]. qPCR was performed using a Bio-Rad CFX96 real-time system with the 5x HOT FIREPol^®^ EvaGreen^®^ qPCR Supermix (Solis BioDyne, Tartu, Estonia) and the following human-specific primer pairs: IL-6: 5′-CAGCTATGAACTCCTTCTCCAC-3′ and 5′-GCGGCTACATCTTTGGAATCT-3′; COX-2: 5’-TACTGGAAGCCAAGCACTTT-3’ and 5’-GGACAGCCCTTCACGTTATT-3’; IL-1b: 5’-CAAAGGCGGCCAGGATATAA-3’ and 5’-CTAGGGATTGAGTCCACATTCAG-3’; Actb: 5′-TTCTACAATGAGCTGCGTGTGGCT-3′ and 5′-TAGCACAGCCTGGATAGCAACGTA-3′. Primers were designed using the Primer Quest Tool software and synthesized by Integrated DNA Technologies Inc. (Montreal, Quebec, Canada). Melting curve analysis was performed to verify amplification specificity. Threshold cycles (Ct) were determined for IL-6, COX-2, IL-1b and Actb, and the relative gene expression was calculated via the 2-(ΔΔCt) method. One-way analysis of variance with repeated measures (ANOVA RM) and Tukey post hoc test was used to compare statistical differences in log2(ΔΔCt) values between treated and LPS samples, as described previously, with a level of significance of * *p* < 0.05, ** *p* < 0.01, and *** *p* < 0.001 [35].

## 3. Results and Discussion

### 3.1. Particle Size and Morphology

SEM was used to visualize the morphology and particle size (PS) of the spray dried samples. The images revealed smooth surfaced, round shaped, spherical particles in all spray dried samples (Figure 3) and the average PS measured by the ImageJ program was between 612 nm and 871 nm (Table 2). 

### 3.2. Thermal Properties 

The DSC curves of the PMs and the spray dried samples are shown in Figure 4. The endothermic peaks at around 170 °C in the PMs are the melting points of raw MEL-P indicating its crystallinity [36]. The broad endothermic peaks from 40 °C to 105 °C in Figure 4a, and the endothermic bands from 60 °C to 105 °C shown in Figure 4b in the curves of the PMs are due to the dehydration of HPβCD and αCD, respectively [37,38,39]. In the case of the spray dried samples, besides the broad endothermic peaks caused by the loss of water, no thermal event could be observed. The disappearance of the endothermic peaks of MEL-P can suggest not only its amorphization, but the formation of inclusion complexes [40]. However, above 225 °C, the appearance of exotermic peaks in the PMs and in the products are due to the decomposition of MEL-P. 

### 3.3. Structural Characterization

The XRPD diffractograms of the spray dried samples and the PMs are shown in Figure 5. The distinct peaks appearing at 6.1, 15.5, 24.6 and 30.9 2Θ values indicate the crystallinity of raw MEL-P in the PMs. In Figure 5a, the absence of characteristic peaks for HPβCD in the PMs suggesting its amorphous state. and in Figure 5b, the characteristic peaks at diffraction angles 2Θ of 5.3°, 12.0°, 14.4° and 21.8° indicate the crystallinity of αCD [41]. After spray drying, the intensity of the previously mentioned diffraction peaks assigned to MEL-P and αCD remarkably reduced referring to their amorphization and the formation of the inclusion complexes. These results corresponded to those of the thermal analysis. No change could be observed in the crystallinity of the products after three months of storage in laboratory conditions.

### 3.4. Secondary Interactions

The FT-IR spectra of raw HPβCD showed characteristic peaks at 1653 cm^−^^1^ (H–O–H bending), 2931 cm^−^^1^ (C–H stretching) and 3404 cm^−^^1^ (O–H stretching) [42]. In all of the HPβCD-based spray dried samples, the bands shifted to lower wavenumbers: 1653 cm^−^^1^ to 1616 cm^−^^1^ and 3404 cm^−^^1^ to 3385 cm^−^^1^ (Figure 6a). Considering raw αCD, the band at 3405 cm^−^^1^ shifted to lower wavenumbers (3385 cm^−^^1^) as well as in the αCD-based spray dried samples. In addition, wavenumber of band of αCD decreased from 1643 cm^−^^1^ to 1616 cm^−^^1^, as well. These peaks are assigned to the stretching and the bending vibration of H-O in αCD, respectively (Figure 6b). The characteristic bands of HA and PVA were probably shaded in the spray dried samples by the other compounds. These changes can indicate the formation of hydrogen bonds between the MEL-P and the cyclodextrins.

### 3.5. Mucoadhesivity

The influence of HA and PVA on the potential mucoadhesivity of the samples was estimated. As controls, pure agar gels were applied, and all of the spray dried samples moved to the bottom of the petri dishes in the first minute of the investigation. In the case of the αCD-containing samples, the displacement of αCD_MEL-P_HA_spd was higher (3.3 cm) than the αCD_MEL-P_PVA_spd (2.69 cm), in 20 min. As an effect of the osmotic activity of HPβCD, HPβCD_MEL-P_HA_spd and HPβCD_MEL-P_PVA_spd samples adsorbed water from the gels resulting in their dissolution and complete displacement towards the bottom of the petri dishes in the first two minutes of the investigation. However, the PVA-containing sample moved slower. These results indicate higher mucoadhesivity of PVA compared to HA in the formulations. 

### 3.6. In Vitro and Ex Vivo Permeability 

For the in vitro test, the cumulative amount of MEL-P that diffused through the artificial membrane was measured as a function of time in a modified horizontal diffusion cell (Figure 7). The flux at 15 min (Figure 8) and the enhancement ratios (Table 3) were determined. In the case of the HPβCD-based samples (Figure 7a), the highest amount of MEL-P permeated from the PVA-containing sample, where 194 μg/cm^2^ of MEL-P diffused to the acceptor phase in 60 min. According to the enhancement ratio values, 7 times more drug permeated from this formulation, than of raw MEL-P. The same tendency could be observed in the case of the αCD-based samples (Figure 7b). The highest amount—209 μg/cm^2^ in 60 min—of MEL-P permeated from the PVA-containing sample as well, more than 7.5 times more API could diffuse through the membrane. In both cases, the presence of PVA seemed to have a beneficial effect on the permeated amount of the drug. The same was experienced by Kaur, Indu P. et al., where the in vitro corneal permeation of acetazolamide was outstanding from their HPβCD-PVA-containing formulation [43]. According to literature data, this phenomenon presumably can be explained by the increasing effect of PVA on the free drug concentration in the aqueous diffusion layer on the surface of the biological and artificial membranes [43,44].

In contrast, the in vitro permeation enhancing effect of HA was not clearly convincing. Although the presence of HA was favorable in the HPβCD-based sample, in the case of the αCD-based sample, according to the enhancement ratio values, less amount of MEL-P diffused to the acceptor phase compared to the polymer-free formulation. In αCD_MEL-P_HA_spd, the deterioration of diffusion could occur due to the hindering effect of HA swelling retaining the release of MEL-P. 

All of the prepared formulations provided higher in vitro permeation of the API than raw MEL-P; the use of PVA in the products seemed to have a distinctly beneficial effect on the amount of drug diffused under nasal conditions. 

The flux at 15 min followed a similar tendency as the permeation sequence (Figure 8). The αCD_MEL-P_PVA_spd tended to diffuse the quickest through the membrane in vitro.

For the ex vivo measurements, the two PVA-containing samples, HPβCD_MEL-P_PVA_spd and αCD_MEL-P_PVA_spd were tested (Figure 9), because they showed the best in vitro results. The highest amount of MEL-P permeated from the αCD-based sample, where 1.47 μg/mm^2^ MEL-P diffused into the acceptor phase in 60 min. From the HPβCD_MEL-P_PVA_spd sample, only less than a third of the aforementioned amount of MEL-P, 0.45 μg/mm^2^, permeated to the acceptor phase in 60 min. This phenomenon can be corresponding to the potential higher permeability enhancing effect of αCD than HPβCD by interacting with membrane phospholipids in the human nasal mucosal cells [45]. These results corresponded to those of the in vitro measurements. 

### 3.7. In Vitro Cytotoxicity and IL-6, COX-2, IL-1b Expression

Cytotoxicity measurement revealed that the noncytotoxic concentrations are 1.03 mg/mL HPβCD_MEL-P_PVA_spd and 0.38 mg/mL αCD_MEL-P_PVA_spd. These results corresponded to the literature data, where a higher tolerable concentration was detected for HPβCD than αCD [46]. LPS was not cytotoxic to the cells.

RPMI 2650 cells were treated with 1.03 µg/mL HPβCD_MEL-P_PVA_spd and 5 µg/mL LPS or 0.38 µg/mL αCD_MEL-P_PVA_spd solution and 5 µg/mL LPS or 5 µg/mL LPS or left untreated. Cells were collected 24 h post-treatment; RNA was extracted, and RT-qPCR was conducted to check IL-6, COX-2 and IL-1b relative expression. The bar denotes the mean and the standard deviation of the expression levels for triplicate measurements. (* *p* < 0.05, ** *p* < 0.01, and *** *p* < 0.001).

Furthermore, we wanted to check the potential anti-inflammatory effect of the compounds. LPS significantly elevated IL-6 relative expression compared to the untreated group 1.71-fold (Figure 10). LPS also elevated COX-2 (Figure 11) and IL-1b (Figure 12) relative expression 1.56-fold and 1.585-fold, respectively. All the examined compounds significantly decreased IL-6, COX-2 and IL-1b relative expression compared to LPS, HPβCD_MEL-P_PVA_spd 0.277-fold, 0.28-fold and 0.01-fold, respectively and αCD_MEL-P_PVA_spd 0.307-fold, 0.16-fold, and 0.02-fold, respectively.

## 4. Conclusions

In this work, MEL-P-containing particles were prepared by nano spray drying using HPβCD and αCD as permeability enhancers, and HA and PVA as mucoadhesive excipients to obtain appropriate formulations for nasal administration. Using these additives, MEL-P-cyclodextrin and MEL-P-cyclodextrin-polymer nanospheres were formulated to study the effect of the type of cyclodextrin and the polymer in the composition on the performance of the samples. The physico-chemical characterization, mucoadhesivity test, in vitro permeability and cytotoxicity and ex vivo permeability studies were carried out. In all cases, nanospheres were successfully prepared (average PS <1 µm) while MEL-P was present in mostly an amorphous state confirmed by DSC and XRPD. Secondary interactions were formed between the API and the cyclodextrins in each product, indicating the complexation of MEL-P. The PVA-containing samples showed higher mucoadhesivity than HA-containing ones, and among all the products, αCD_MEL-P_PVA_spd had the potential highest mucoadhesive property according to our results. The in vitro flux at 15 min was higher from HPβCD-containing samples than from αCD-containing ones, and except for the PVA-containing products, the permeation extent from the αCD_MEL-P_PVA_spd was the highest (209 μg/cm^2^ in 60 min) among the samples. The presence of HA resulted in the decrease of in vitro permeation of MEL-P due to the retaining effect of HA swelling, and meanwhile the presence of PVA caused remarkable increase of the permeation rate in the case of both cyclodextrins in vitro. The difference between the αCD_MEL-P_PVA_spd and the HPβCD_MEL-P_PVA_spd samples was more noticeable ex vivo (1.47 μg/mm^2^ and 0.45 μg/mm^2^, respectively), which can be due to the higher permeation enhancing feature of αCD compared to HPβCD on nasal epithelial cells; however, HPβCD_MEL-P_PVA_spd had a higher tolerable concentration according to the cytotoxicity measurement. The two aforementioned formulations showed significant (*p* < 0.001) anti-inflammatory effect. Overall, αCD_MEL-P_PVA_spd showed the best results of all the products based on our measurements. 

The prepared formulations may be suitable for rapid onset of analgesic effect or as adjuvants to opioids through the nasal route.

## Figures and Tables

**Figure 1 pharmaceutics-13-01883-f001:**
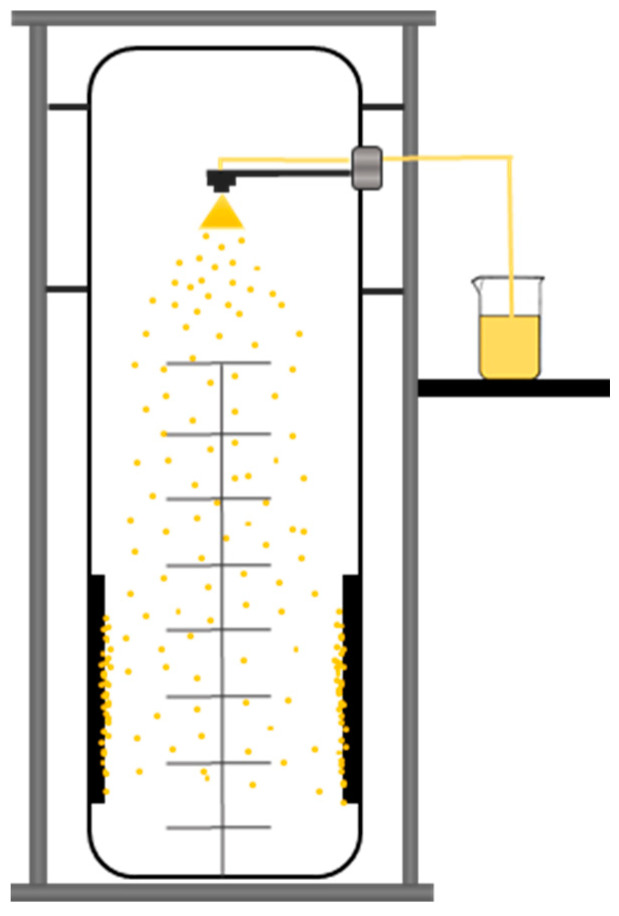
Schematic image of BÜCHI Nano Spray Dryer B-90 HP.

**Figure 2 pharmaceutics-13-01883-f002:**
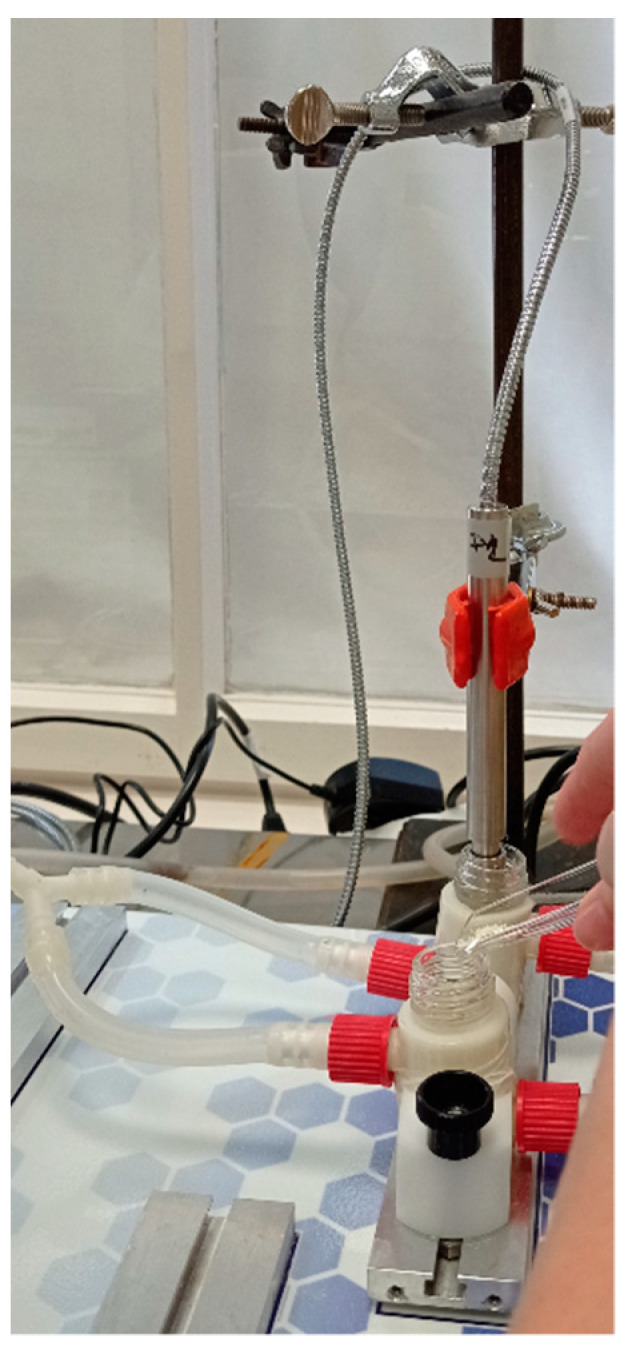
Modified horizontal diffusion model used for the in vitro and ex vivo permeability studies [33].

**Figure 3 pharmaceutics-13-01883-f003:**
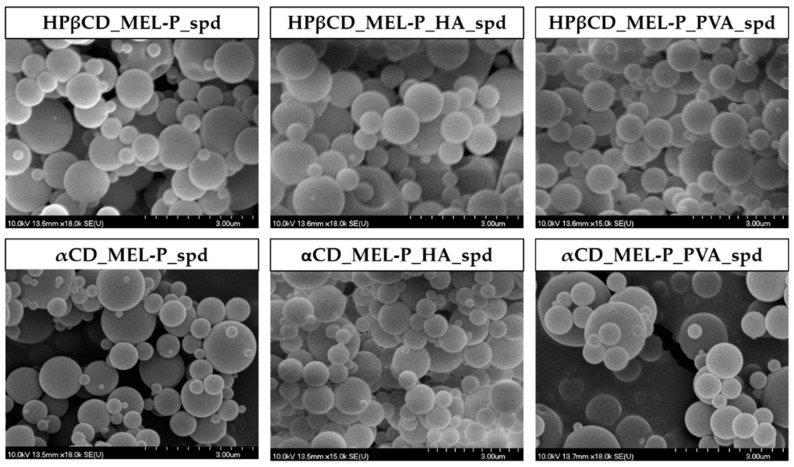
SEM images of the spray dried samples. The scale bar = 3.0 µm.

**Figure 4 pharmaceutics-13-01883-f004:**
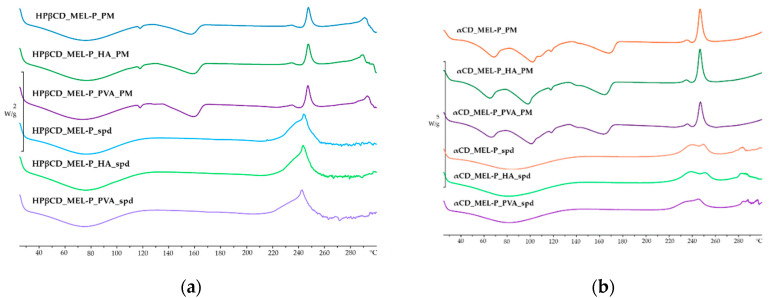
DSC curves of: (**a**) the HPβCD-containing and (**b**) the αCD-containing PMs and spray dried samples.

**Figure 5 pharmaceutics-13-01883-f005:**
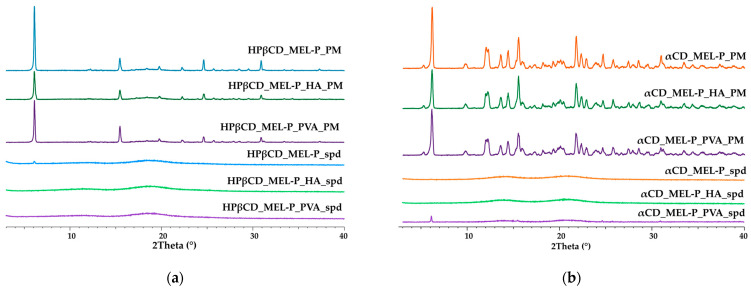
XRPD diffractograms of: (**a**) the HPβCD-containing and (**b**) the αCD-containing PMs and spray dried samples.

**Figure 6 pharmaceutics-13-01883-f006:**
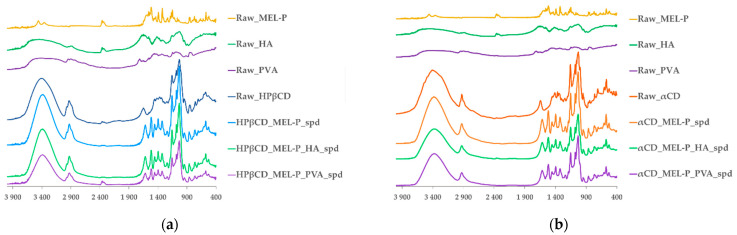
FT-IR spectra of the raw materials and: (**a**) the HPβCD-containing and (**b**) the αCD-containing spray dried samples.

**Figure 7 pharmaceutics-13-01883-f007:**
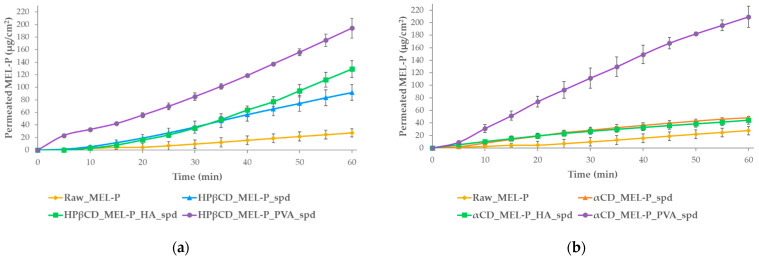
In vitro permeability study of raw MEL-P and: (**a**) the HPβCD-containing and (**b**) the αCD-containing spray dried samples.

**Figure 8 pharmaceutics-13-01883-f008:**
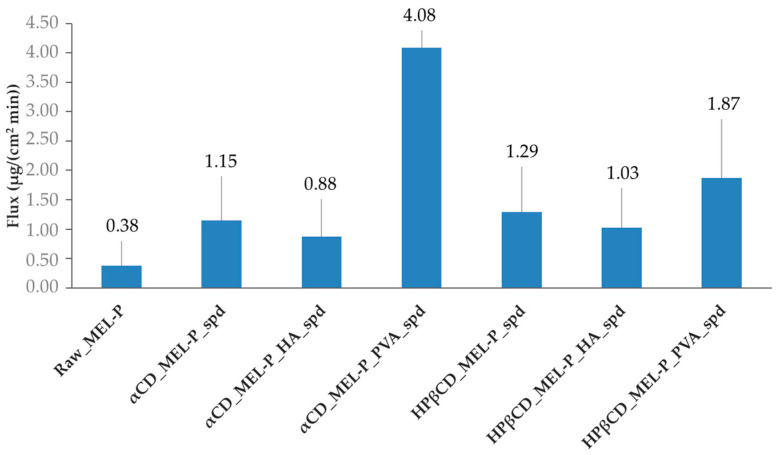
Flux of the spray dried samples at 15 min.

**Figure 9 pharmaceutics-13-01883-f009:**
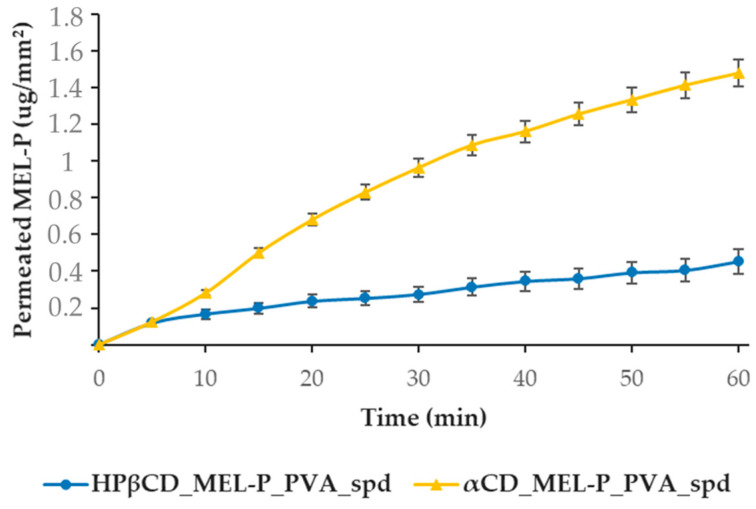
Ex vivo permeability study of HPβCD_MEL-P_PVA_spd and αCD_MEL-P_PVA_spd on human nasal mucosa.

**Figure 10 pharmaceutics-13-01883-f010:**
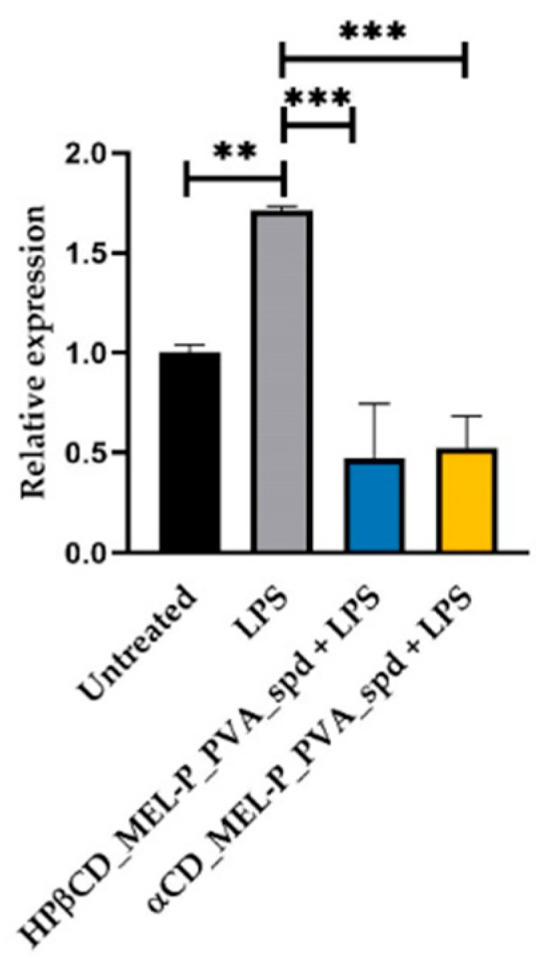
Relative expression of *IL-6*. (level of significance: ** *p* < 0.01, and *** *p* < 0.001).

**Figure 11 pharmaceutics-13-01883-f011:**
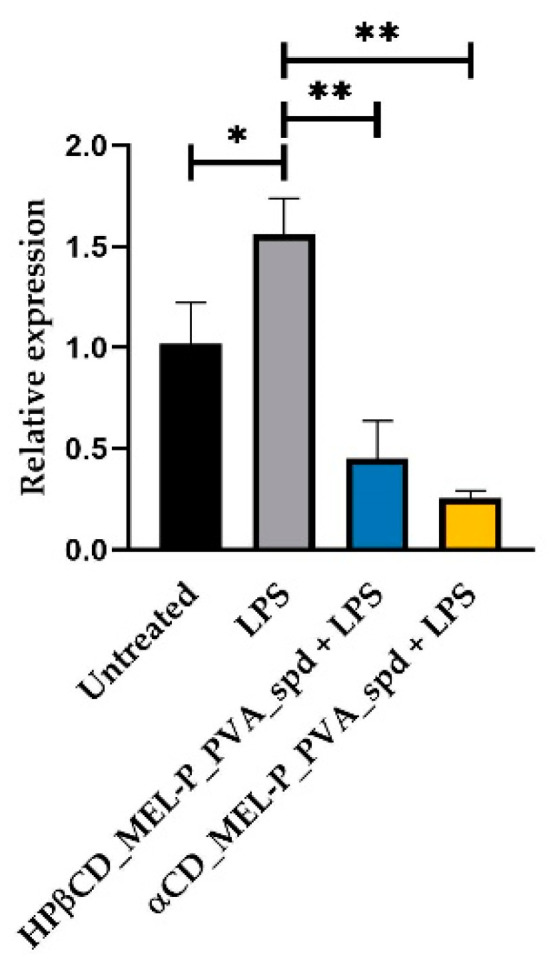
Relative expression of *COX-2*. (level of significance: * *p* < 0.05, ** *p* < 0.01).

**Figure 12 pharmaceutics-13-01883-f012:**
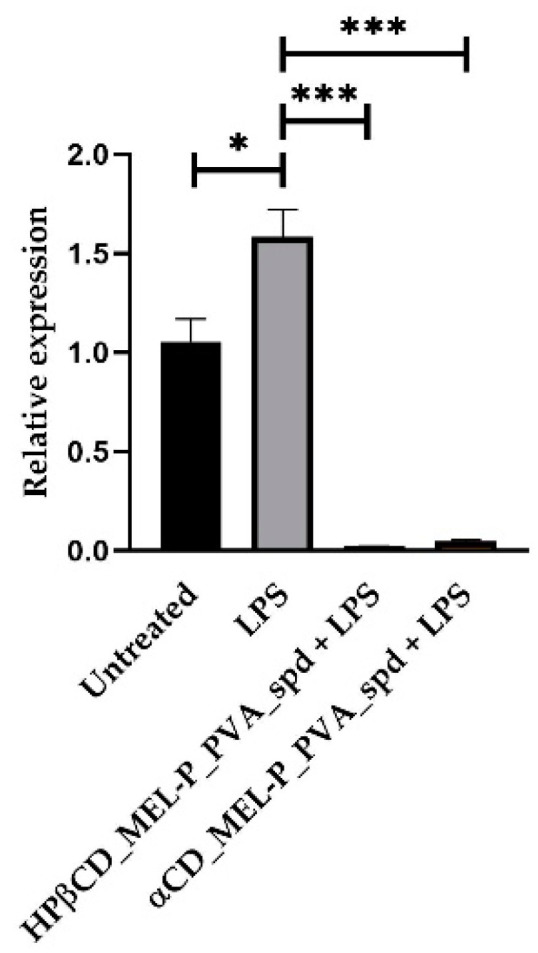
Relative expression of *IL-1b*. (level of significance: * *p* < 0.05, *** *p* < 0.001).

**Table 1 pharmaceutics-13-01883-t001:** Composition of the samples.

Samples	Distilled Water (mL)	HPβCD (mg)	αCD (mg)	HA (mg)	PVA (mg)	MEL-P (mg)
HPβCD_MEL-P	10	264.79	-	-	-	70
HPβCD_MEL-P_HA	10	264.79	-	5	-	70
HPβCD_MEL-P_PVA	10	264.79	-	-	10	70
αCD_MEL-P	10	-	167.11	-	-	70
αCD_MEL-P_HA	10	-	167.11	5	-	70
αCD_MEL-P_PVA	10	-	167.11	-	10	70

**Table 2 pharmaceutics-13-01883-t002:** Average particle size (PS).

Composition	Avarage PS (nm)
HPβCD_MEL-P_spd	871 ± 439
HPβCD_MEL-P_HA_spd	868 ± 243
HPβCD_MEL-P_PVA_spd	723 ± 229
αCD_MEL-P_spd	612 ± 227
αCD_MEL-P_HA_spd	756 ± 175
αCD_MEL-P_PVA_spd	799 ± 256

**Table 3 pharmaceutics-13-01883-t003:** Enhancement ratios.

Formulation	Enhancement Ratio
HPβCD_MEL-P_spd	3.33
HPβCD_MEL-P_HA_spd	4.68
HPβCD_MEL-P_PVA_spd	7.05
αCD_MEL-P_spd	1.75
αCD_MEL-P_HA_spd	1.61
αCD_MEL-P_PVA_spd	7.60

## Data Availability

The data presented in this study are available on request from the corresponding author.

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
