# Peer review of "Physico-Chemical, In Vitro and Ex Vivo Characterization of Meloxicam Potassium-Cyclodextrin Nanospheres"

_pharmaceutics, 2021, doi:10.3390/pharmaceutics13111883_

Round 1
Reviewer 1 Report
- What is the significance of using soluble drug with cyclodextrinand then incorpoarting through nasal routes.
- How much received after formulation of spray dried powder.
- What was the size found for prepared deliver. ADmibnistration of drug through nasal route is very critical.
- In section 3.1, size was mentined in nanometer and Table written micrometer.
- Among used CDs, Which one was the best composition.
- Give the release graph.
- In abstract give the important numerical value of result.
Author Response
Reply to Referee comments
For Reviewer 1
Thank you very much for your remarks. We greatly appreciate your advices. Here you can see listed all of the modifications made in the paper according to your suggestions (shown in blue colour in the text).
- What is the significance of using soluble drug with cyclodextrinand then incorpoarting through nasal routes.
Thank you for your question. The aim of using cyclodextrins was to enhance the permeation of meloxicam potassium through the nasal route. (Papakyriakopoulou, Paraskevi, et al. "Nasal powders of quercetin-β-cyclodextrin derivatives complexes with mannitol/lecithin microparticles for Nose-to-Brain delivery: In vitro and ex vivo evaluation." International Journal of Pharmaceutics 607 (2021): 121016.)
- How much received after formulation of spray dried powder
Thank you for your question. The yield of the formulations was the following:
|
Formulation |
Yield (%) |
|
|
HPβCD_MEL-P_spd |
56.75 |
|
|
HPβCD_MEL-P_HA_ spd |
52.97 |
|
|
HPβCD_MEL-P_PVA_ spd |
75.41 |
|
|
αCD_MEL-P_ spd |
65.20 |
|
|
αCD_MEL-P_HA_ spd |
61.96 |
|
|
αCD_MEL-P_PVA_ spd |
72.86 |
|
- What was the size found for prepared deliver. ADmibnistration of drug through nasal route is very critical.
Thank you for your question. Based on literature data, the preferred particle size for nasal delivery is above 10 m, but it is not defined obviously. However, it was observed that by the administration of particles below 45 m, more uniform and complete covering of the nasal surface can be obtained. In our formulations, PVA and HA were used to provide mucoadhesion, so with the suitable device, the nanoparticles can be delivered to the proper site of the nasal cavity and adhere to the mucosa, therefore the size of the particles is not supposed to be an issue. (Trenkel, Marie, and Regina Scherließ. "Nasal Powder Formulations: In-Vitro characterisation of the impact of powders on nasal residence time and sensory effects." Pharmaceutics 13.3 (2021))
- In section 3.1, size was mentined in nanometer and Table written micrometer.
Thank you for your remark, Table 2 was modified in the manuscript as follows:
Table 2. Avarage particle size (PS)
|
Composition |
Avarage PS (nm) |
|
|
HPβCD_MEL-P_spd |
871 ± 439 |
|
|
HPβCD_MEL-P_HA_ spd |
868 ± 243 |
|
|
HPβCD_MEL-P_PVA_ spd |
723 ± 229 |
|
|
αCD_MEL-P_ spd |
612 ± 227 |
|
|
αCD_MEL-P_HA_ spd |
756 ± 175 |
|
|
αCD_MEL-P_PVA_ spd |
799 ± 256 |
|
- Among used CDs, Which one was the best composition.
Thank you for your question.
According to „5. Conclusions” section:
- „[…] among all the products, αCD_MEL-P_PVA_spd had the potential highest mucoadhesive property according to our results.”
- „[…] the permeation extent from the αCD_MEL-P_PVA_spd was the highest (209 μg/cm2 in 60 min) among the samples.”
- „The difference between the αCD_MEL-P_PVA_spd and the HPβCD_MEL-P_PVA_spd samples was more noticeable ex vivo (1.47 μg/mm2 and 0.45 μg/mm2, respectively) which can be due to the higher permeation enhancing feature of αCD compared to HPβCD on nasal epithelial cells.”
In summary, the αCD_MEL-P_PVA_spd composition showed the best results among all the formulations.
- Give the release graph.
Thank you for your remark. We performed the drug dissolution studies, you can see a release graphs below.
The dissolution tests were carried out in 25 ml of SNES (simulated nasal electrolyte solution) of pH=5.6 at 32 °C applying a modified The European Pharmacopoeia (6th Edition) paddle method. The rotation speed of the paddles was 100 rpm. 3.75 mg MEL-P containing samples were applied, and at predetermined intervals, the amount of dissolved MEL was determined spectrophotometrically at 364 nm.
In case of every HPβCD-based products, we could see an initial burst effect, almost 100% of the MEL-P dissolved in the first 5 minutes. The type of the polymer apparently did not affect the dissolution of the drug. The dissolution of raw MEL-P was not completed during the time of the test, it probably encountered a solubility limit, approximately 50% of it released.
In case of the αCD-based formulations, an initial burst effect could be observed as well for the polymer free and PVA containing samples, more than 90% of MEL-P released in the first 5 minutes. However, HA had a hindering effect on the drug release in αCD_MEL-P_PVA_spd, which is different from the HPβCD_MEL-P_HA_ spd sample. This hindering effect may be due to the higher mass ratio of HA compared to αCD than HPβCD in the formulation, although the released amount of MEL-P is still higher, than that of from raw MEL-P.
- In abstract give the important numerical value of result.
Thank you for your remark, the abstract was modified based on your suggestion.
„[…] We aimed to produce nanoparticles containing meloxicam potassium (MEL-P) by spray drying intended for nasal application. Various cyclodextrins (hydroxypropyl-β-cyclodextrin, α-cyclodextrin) and biocompatible polymers (hyaluronic acid, poly(vinylalcohol)) were used as excipients to increase the permeation of the drug and to prepare mucoadhesive products. Physico-chemical, in vitro and ex vivo biopharmaceutical characterization of the formulations were performed. As a result of spray-drying, mucoadhesive nanospheres (average particle size <1 m) were prepared which contained amorphous MEL-P. Cyclodextrin-MEL-P complexes were formed and the applied excipients increased the in vitro and ex vivo permeability of MEL-P. The highest amount of MEL-P permeated from the α-cyclodextrin-based poly(vinylalcohol)-containing samples in vitro (209 μg/cm2) and ex vivo (1.47 μg/mm2) as well. After further optimization, the resulting formulations may be promising for eliciting a rapid analgesic effect through the nasal route.”
- 10. 2021. Szeged, Hungary Csilla Balla-Bartos PhD
Assistant Professor

Reviewer 2 Report
Review of Manuscript for the Journal of Pharmaceutics
Manuscript ID: pharmaceutics-1383679
Recommendation: It may be accepted for publication with minor revision.
General Comments: This is an interesting work in which the authors prepared mucoadhesive meloxicam potassium-cyclodextrin spray dried nanoparticles for nasal application using hydroxypropyl-β-cyclodextrin, α-cyclodextrin, hyaluronic acid and poly(vinylalcohol)) as excipients to increase the in vitro and ex vivo permeability of meloxicam potassium.
The manuscript is generally well written and the results are clearly presented. Overall it meets the criteria of the Journal and can be published after minor revision.
Some remarks and corrections to be done before publication:
Figure 1. I don’t quite understand the purpose of the photo with the spray dryer instrument. (Maybe a schematic description of the spray-dryer would be more useful to show instead, but I leave that for the authors to decide)
Figure 2. The same modified horizontal diffusion model is published in a previous work of the authors (with the same picture). Perhaps the authors should cite also this work: Bartos, C.; Varga, P.; Szabó-Révész, P.; Ambrus, R. Physico-Chemical and In Vitro Characterization of Chitosan-Based Microspheres Intended for Nasal Administration. Pharmaceutics 2021, 13,608.
Figure 3. Some of the SEM images are in 3 μm scale while others are in 2 μm scale. It would be better for all SEM images to be in the same scale.
2.2.3. Differential scanning calorimetry (DSC). The authors should mention what type of pans they have used. (Was it aluminium pans? There were sealed or not sealed ?)
Line 78-81: “Spray drying is a suitable method for obtaining these complexes, which may have drastically different properties than the original compounds, they can be used to increase the solubility and the dissolution of the APIs. The sentence should be rephrased. (Perhaps to: “Spray drying is a suitable method for obtaining such complexes, which may be used to increase the solubility and the dissolution of the APIs and may have drastically different properties than the original compounds”
Line 101: alpha-cyclodextrin to α-cyclodextrin
Line 239: “..no thermal event could be observed, the disappearance of..” perhaps should change to “..no thermal event could be observed, with the disappearance of..”
Author Response
Reply to Referee comments
For Reviewer 2
Thank you very much for your valuable remarks. We greatly appreciate your advices. Here you can see listed all of the modifications made in the paper according to your suggestions (shown in red colour in the text).
- Figure 1. I don’t quite understand the purpose of the photo with the spray dryer instrument. (Maybe a schematic description of the spray-dryer would be more useful to show instead, but I leave that for the authors to decide)
Thank you for your remark, Figure 1. was replaced as you can see below:
Figure 1. Schematic image of BÜCHI Nano Spray Dryer B-90 HP
- Figure 2. The same modified horizontal diffusion model is published in a previous work of the authors (with the same picture). Perhaps the authors should cite also this work: Bartos, C.; Varga, P.; Szabó-Révész, P.; Ambrus, R. Physico-Chemical and In Vitro Characterization of Chitosan-Based Microspheres Intended for Nasal Administration. Pharmaceutics2021, 13,608.
Thank you for your suggestion, the article was cited.
Figure 2. Modified horizontal diffusion model used for the in vitro and ex vivo permeability studies [33]
[33] Bartos, C.; Varga, P.; Szabó-Révész, P.; Ambrus, R. Physico-Chemical and In Vitro Characterization of Chitosan-Based Microspheres Intended for Nasal Administration. Pharmaceutics 2021, 13,608.
- Figure 3. Some of the SEM images are in 3 μm scale while others are in 2 μm scale. It would be better for all SEM images to be in the same scale.
Thank you for your remark, the images in 2 μm scale were replaced in Figure 3.
Figure 3. SEM images of the spray dried samples
- 2.2.3. Differential scanning calorimetry (DSC). The authors should mention what type of pans they have used. (Was it aluminium pans? There were sealed or not sealed ?)
Thank you for your suggestion. The text was modified as follows:
“2.2.3. Differential scanning calorimetry (DSC)
Mettler Toledo DSC 821e (Germany) system and STARe program V9.1 (Mettler Inc., Schwerzenbach, Switzerland) were used to implement the thermal analysis. Approximately 2–5 mg of samples in sealed aluminium pans were heated from 25 °C to 300 °C applying 10 °C·min−1 heating rate under a constant argon flow of 10 l·h−1. Physical mixtures of MEL-P, cyclodextrins, HA and PVA in the same ratio as the spray-dried samples contained were mixed in a Turbula mixer (Turbula WAB, Systems Schatz, Switzerland) at 50 rpm for 10 minutes and were applied as control samples.”
- Line 78-81: “Spray drying is a suitable method for obtaining these complexes, which may have drastically different properties than the original compounds, they can be used to increase the solubility and the dissolution of the APIs. The sentence should be rephrased. (Perhaps to: “Spray drying is a suitable method for obtaining such complexes, which may be used to increase the solubility and the dissolution of the APIs and may have drastically different properties than the original compounds”)
Thank you for your remark. The sentence was rephrased as you suggested.
Line 78-81:
“Spray drying is a suitable method for obtaining such complexes, which may be used to increase the solubility and the dissolution of the APIs and may have drastically different properties than the original compounds.”
- Line 101: alpha-cyclodextrin to α-cyclodextrin
Thank you for your remark, the text was modified according to your suggestion.
Line 104:
„(2-Hydroxy)-propyl-β-cyclodextrin (HPβCD) and α-cyclodextrin (αCD) were from Cyclolab Ltd. (Budapest, Hungary).”
- Line 239: “..no thermal event could be observed, the disappearance of..” perhaps should change to “..no thermal event could be observed, with the disappearance of..”
Thank you for your suggestion, however the sentence was modified slightly different, because we made further findings based on the literature data.
3.2. Thermal properties
“ [...] In case of the spray dried samples, besides the broad endothermic peaks caused by the loss of water, no thermal event could be observed. The disappearance of the endothermic peaks of MEL-P can suggest not only its amorphization, but the formation of inclusion complexes [40].”
[40] Ficarra, R.; Tommasini, S.; Raneri, D.; Calabrò, M.L.; Di Bella, M.R.; Rustichelli, C.; Gamberini, M.C.; Ficarra, P. Study of Flavonoids/β-Cyclodextrins Inclusion Complexes by NMR, FT-IR, DSC, X-Ray Investigation. Journal of Pharmaceutical and Biomedical Analysis 2002, 29, 1005–1014, doi:10.1016/S0731-7085(02)00141-3.
- 10. 2021. Szeged, Hungary Csilla Balla-Bartos PhD
Assistant Professor

Reviewer 3 Report
The work described in the present manuscript is consistent with the scope of the journal. Physico-chemical, in vitro and ex vivo characterization of meloxicam potassium-cyclodextrin nanospheres has discussed in detail the in vitro and ex vivo evaluation of the nanosphere. This work is methodically carried out and scientifically correct. There are major issues that the authors can address to improve their manuscript before acceptance for publication.
In the introduction section, recent literature report on meloxicam is missing need to be included
The authors should calculate enhancement ratio and lag time for permeation studies if possible.
The authors can include some more inflammatory markers such as Cox-2, Tnfα, IL-1β in publishing high impact journal
Discussion of the results is laconic; the author needs to improve the discussion compared to previous literature.
Please indicate both the manufacturer’s name and location (including city, state, and country) for all specialized equipments, kits, software, incubators, instruments, and reagents used in the experiment wherever required
Please provide high resolution (300 dpi) images.
Author Response
Reply to Referee comments
For Reviewer 3
Thank you very much for your valuable remarks. We greatly appreciate your advices. Here you can see listed all of the modifications made in the paper according to your suggestions (shown in green colour in the text).
- In the introduction section, recent literature report on meloxicam is missing need to be included
Thank you for your remark, the introduction section was modified.
„ […] Meloxicam (MEL) is a non-steroidal anti-inflammatory drug with poor water solubility. In the therapeutic field, it is used to treat different joint diseases and it could be used to relive acute pain [23]. Its effect on the use of opioids in case of post-orthopedic surgery patients was studied administering it intravenously [24]. Also, its delivery through alternative routes – such as transdermal and nasal routes – has been recently investigated in vitro and in vivo [25,26]. Meloxicam potassium monohydrate (MEL-P) is the salt form of MEL that has been developed by Egis Ltd. (Hungary) and has a higher aqueous solubility (MEL-P: 13.1 mg/ml in water at 25 °C; MEL: 4.4 μg/mL in water at 25 °C). However, with different techniques – e.g. spray drying, forming inclusion complexes, incorporating into polymer matrices – its permeability and bioavailability can be further increased [27,28].”
[23] Bartos, C.; Ambrus, R.; Sipos, P.; Budai-Szűcs, M.; Csányi, E.; Gáspár, R.; Márki, Á.; Seres, A.B.; Sztojkov-Ivanov, A.; Horváth, T.; et al. Study of Sodium Hyaluronate-Based Intranasal Formulations Containing Micro- or Nanosized Meloxicam Particles. International Journal of Pharmaceutics 2015, 491, 198–207, doi:10.1016/j.ijpharm.2015.06.046.
[24] Sharpe, K.P.; Berkowitz, R.; Tyndall, W.A.; Boyer, D.; McCallum, S.W.; Mack, R.J.; Du, W. Safety, Tolerability, and Effect on Opioid Use of Meloxicam IV Following Orthopedic Surgery. J Pain Res 2020, 13, 221–229, doi:10.2147/JPR.S216219
[25] Kuznetsova, D.A.; Vasileva, L.A.; Gaynanova, G.A.; Vasilieva, E.A.; Lenina, O.A.; Nizameev, I.R.; Kadirov, M.K.; Petrov, K.A.; Zakharova, L.Ya.; Sinyashin, O.G. Cationic Liposomes Mediated Transdermal Delivery of Meloxicam and Ketoprofen: Optimization of the Composition, in Vitro and in Vivo Assessment of Efficiency. International Journal of Pharmaceutics 2021, 605, 120803, doi:10.1016/j.ijpharm.2021.120803
[26] Bartos, C.; Ambrus, R.; Kovács, A.; Gáspár, R.; Sztojkov-Ivanov, A.; Márki, Á.; Janáky, T.; Tömösi, F.; Kecskeméti, G.; Szabó-Révész, P. Investigation of Absorption Routes of Meloxicam and Its Salt Form from Intranasal Delivery Systems. Molecules 2018, 23, 784, doi:10.3390/molecules23040784
[27] Castilla-Casadiego, D.A.; Carlton, H.; Gonzalez-Nino, D.; Miranda-Muñoz, K.A.; Daneshpour, R.; Huitink, D.; Prinz, G.; Powell, J.; Greenlee, L.; Almodovar, J. Design, Characterization, and Modeling of a Chitosan Microneedle Patch for Transdermal Delivery of Meloxicam as a Pain Management Strategy for Use in Cattle. Materials Science and Engineering: C 2021, 118, 111544, doi:10.1016/j.msec.2020.111544
[28] Chvatal, A.; Farkas, Á.; Balásházy, I.; Szabó-Révész, P.; Ambrus, R. Aerodynamic Properties and in Silico Deposition of Meloxicam Potassium Incorporated in a Carrier-Free DPI Pulmonary System. International Journal of Pharmaceutics 2017, 520, 70–78, doi:10.1016/j.ijpharm.2017.01.070
- The authors should calculate enhancement ratio and lag time for permeation studies if possible.
Thank you for your suggestions. Unfortunately, the lag time could not be clearly determined from our measurement results, however the enhancement ratio was calculated and the following modifications were carried out in the manuscript according to it:
„2.2.7. In vitro and ex vivo permeability studies
[…] Each measurement was carried out in triplicate. The flux was determined at 15 min and the permeation enhancement ratios for the in vitro measurements were calculated based on the following equations (Equation (1) and Equation (2)) [32]:
|
(1) |
where Papp is the apparent permeability coefficient (cm/s), Q is the total amount permeated throughout the incubation time (g), A is the diffusion area of the artificial membrane (cm2), c is the initial concentration of the drug in the donor compartment (g/cm3), and t is the total time of the experiment (s) and
|
(2) |
where R is the permeation enhancement ratio and Papp(control) is the apparent permeability coefficient (cm/s) of MEL-P.”
“3.6. In vitro and ex vivo permeability
For the in vitro test, the cumulative amount of MEL-P that diffused through the artificial membrane was measured as a function of time in a modified horizontal diffusion cell. The flux at 15 min (Figure 8.) and the enhancement ratios (Table 3.) were determined. In the case of the HPβCD-based samples (Figure 7. a), the highest amount of MEL-P permeated from the PVA containing sample, where 194 μg/cm2 of MEL-P diffused to the acceptor phase in 60 min. According to the enhancement ratio values, 7 times more drug permeated from this formulation, than of raw MEL-P. The same tendency could be observed in case of the αCD-based samples (Figure 7. b). The highest amount – 209 μg/cm2 in 60 min – of MEL-P permeated from the PVA-containing sample as well, more than 7.5 times more API could diffuse through the membrane. […] Although, the presence of HA was favourable in the HPβCD-based sample, in case of the αCD-based sample, according to the enhancement ratio values, less amount of MEL-P diffused to the acceptor phase compared to the polymer free formulation.”
Table 3. Enhancement ratios
|
Formulation |
Enhancement ratio |
|
|
HPβCD_MEL-P_spd |
3.33 |
|
|
HPβCD_MEL-P_HA_ spd |
4.68 |
|
|
HPβCD_MEL-P_PVA_ spd |
7.05 |
|
|
αCD_MEL-P_ spd |
1.75 |
|
|
αCD_MEL-P_HA_ spd |
1.61 |
|
|
αCD_MEL-P_PVA_ spd |
7.60 |
|
[32] Clausen, A. E., Kast, C. E., & Bernkop-Schnürch, A. (2002). The role of glutathione in the permeation enhancing effect of thiolated polymers. Pharmaceutical research, 19(5), 602-608.
- The authors can include some more inflammatory markers such as Cox-2, Tnfα, IL-1β in publishing high impact journal.
Thank you for your suggestion.
We performed COX-2 and IL-1b RT-qPCR measurements, but the TNF-α gene expression rate of RPMI 2650 epithelial cells is very low, so it could not be evaluated.
The following modifications were carried out in the manuscript according to the performed measurements:
„2.2.11. qPCR amplification of IL-6, COX-2, IL-1b, Actb
qPCR was performed using a Bio-Rad CFX96 real-time system with the 5x HOT FIREPol® EvaGreen® qPCR Supermix (Solis BioDyne, Tartu, Estonia) and the following human-specific primer pairs: IL-6: 5’-CAGCTATGAACTCCTTCTCCAC-3’, and 5’-GCGGCTACATCTTTGGAATCT -3’; COX-2: 5'-TACTGGAAGCCAAGCACTTT-3' and 5'-GGACAGCCCTTCACGTTATT-3'; IL-1b: 5'-CAAAGGCGGCCAGGATATAA-3' and 5'-CTAGGGATTGAGTCCACATTCAG-3'; Actb: 5’-TTCTACAATGAGCTGCGTGTGGCT-3’, and 5’-TAGCACAGCCTGGATAGCAACGTA -3’. […]”
“3.7. In vitro cytotoxicity and IL-6, COX-2, IL-1b expression
[…] Furthermore, we wanted to check the potential anti-inflammatory effect of the compounds. LPS significantly elevated IL-6 relative expression compared to Untreated group 1.71-fold (Figure 10.). LPS also elevated COX-2 (Figure 11.) and IL-1b (Figure 12.) relative expression 1.56-fold and 1.585-fold, respectively. All the examined compounds significantly decreased IL-6, COX-2 and IL-1b relative expression compared to LPS, HPβCD_MEL-P_PVA_spd 0.277-fold, 0.28-fold and 0.01-fold, respectively and αCD_MEL-P_PVA_spd 0.307-fold, 0.16-fold, 0.02-fold, respectively.”
Figure 11. Relative expression of COX-2
Figure 12. Relative expression of IL-1b
- Discussion of the results is laconic; the author needs to improve the discussion compared to previous literature.
Thank you for your remark, the following modifications were carried out in the „3. Results and discussion” part according to your comment:
„3.2. Thermal properties
[…] In case of the spray dried samples, besides the broad endothermic peaks caused by the loss of water, no thermal event could be observed. The disappearance of the endothermic peaks of MEL-P can suggest not only its amorphization, but the formation of inclusion complexes [40]. […]” – It is shown in red colour in the manuscript because Rewiever 2 suggested also some modification in this part.
„3.6. In vitro and ex vivo permeabilty
For the in vitro test, the cumulative amount of MEL-P that diffused through the artificial membrane was measured as a function of time in a modified horizontal diffusion cell. The flux at 15 min (Figure 8.) and the enhancement ratios (Table 3.) were determined. In the case of the HPβCD-based samples (Figure 7. a), the highest amount of MEL-P permeated from the PVA containing sample, where 194 μg/cm2 of MEL-P diffused to the acceptor phase in 60 min. According to the enhancement ratio values, 7 times more drug permeated from this formulation, than of raw MEL-P. The same tendency could be observed in case of the αCD-based samples (Figure 7. b). The highest amount – 209 μg/cm2 in 60 min – of MEL-P permeated from the PVA-containing sample as well, more than 7.5 times more API could diffuse through the membrane. In both cases, the presence of PVA seemed to have a beneficial effect on the permeated amount of the drug. The same was experienced by Kaur, Indu P., et al., where the in vitro corneal permeation of acetazolamide was outstanding from their HPβCD-PVA containing formulation [43]. According to literature data, this phenomenon presumably can be explained by the increasing effect of PVA on the free drug concentration in the aquaeous diffusion layer on the surface of the biological and artificial membranes [43,44].
In contrast, the in vitro permeation enhancing effect of HA was not clearly convincing. Although, the presence of HA was favourable in the HPβCD-based sample, in case of the αCD-based sample, according to the enhancement ratio values, less amount of MEL-P diffused to the acceptor phase compared to the polymer free formulation. In αCD_MEL-P_HA_ spd, the deterioration of diffusion could occur due to the hindering effect of HA swelling retaining the release of MEL-P.
All of the prepared formulations provided higher in vitro permeation of the API, than raw MEL-P, the use of PVA in the products seemed to have a distinctly beneficial effect on the amount of drug diffused under nasal conditions. […]
From the HPβCD_MEL-P_PVA_spd sample, only less than the third of the afforementioned amount of MEL-P, 0.45 μg/mm2 permeated to the acceptor phase in 60 min. This phenomenon can be corresponding to the potential higher permeability enhancing effect of αCD than HPβCD by interacting with membrane phospholipids in the human nasal mucosal cells [45]. […]”
„3.7. In vitro cytotoxicity and IL-6, COX-2, IL-1b expression
Cytotoxicity measurement revealed that the noncytotoxic concentrations are 1.03 mg/ml HPβCD_MEL-P_PVA_spd and 0.38 mg/ml αCD_MEL-P_PVA_spd. These results corresponded to the literature data, where a higher tolerable concentration was detected for HPβCD than αCD [46]. […]”
[40] Ficarra, R.; Tommasini, S.; Raneri, D.; Calabrò, M.L.; Di Bella, M.R.; Rustichelli, C.; Gamberini, M.C.; Ficarra, P. Study of Flavonoids/β-Cyclodextrins Inclusion Complexes by NMR, FT-IR, DSC, X-Ray Investigation. Journal of Pharmaceutical and Biomedical Analysis 2002, 29, 1005–1014, doi:10.1016/S0731-7085(02)00141-3
[43] Kaur, I.; Kapil, M.; Smitha, R.; Aggarwal, D. Development of Topically Effective Formulations of Acetazolamide Using HP-β-CD-Polymer Co-Complexes. CDD 2004, 1, 65–72, doi:10.2174/1567201043480054.
[44] Loftsson, T.; Masson, M. The Effects of Water-Soluble Polymers on Cyclodextrins and Cyclodextrin Solubilization of Drugs. Journal of Drug Delivery Science and Technology 2004, 14, 35–43, doi:10.1016/S1773-2247(04)50003-5.
[45] Haimhoffer, Á.; Rusznyák, Á.; Réti-Nagy, K.; Vasvári, G.; Váradi, J.; Vecsernyés, M.; Bácskay, I.; Fehér, P.; Ujhelyi, Z.; Fenyvesi, F. Cyclodextrins in Drug Delivery Systems and Their Effects on Biological Barriers. Sci. Pharm. 2019, 87, 33, doi:10.3390/scipharm87040033.
[46] Rassu, G.; Sorrenti, M.; Catenacci, L.; Pavan, B.; Ferraro, L.; Gavini, E.; Bonferoni, M.C.; Giunchedi, P.; Dalpiaz, A. Versatile Nasal Application of Cyclodextrins: Excipients and/or Actives? Pharmaceutics 2021, 13, 1180, doi:10.3390/pharmaceutics13081180
- Please indicate both the manufacturer’s name and location (including city, state, and country) for all specialized equipments, kits, software, incubators, instruments, and reagents used in the experiment wherever required
Thank you for your remark, the manufacturers’ name and location were checked and corrected.
- Please provide high resolution (300 dpi) images.
Thank you for your remark, the resolution of the images is corrected.
- 10. 2021. Szeged, Hungary Csilla Balla-Bartos PhD
Assistant Professor

Round 2
Reviewer 1 Report
Accept
Reviewer 3 Report
The Authors have responded to all comment raised by reviewer approperiately. The manuscript can be accepted in its current form.